# Invariant Causal Imitation Learning for Generalizable Policies

**Ioana Bica**[*]
University of Oxford, Oxford, UK
The Alan Turing Institute, London, UK
ioana.bica@eng.ox.ac.uk

**Daniel Jarrett**[*]
University of Cambridge, Cambridge, UK
daniel.jarrett@maths.cam.ac.uk

**Mihaela van der Schaar**
University of Cambridge, Cambridge, UK
University of California, Los Angeles, USA
The Alan Turing Institute, London, UK
mv472@cam.ac.uk

## Abstract

Consider learning an imitation policy on the basis of demonstrated behavior from multiple environments, with an eye towards deployment in an unseen environment. Since the observable features from each setting may be different, directly learning individual policies as mappings from features to actions is prone to *spurious correlations*—and may not generalize well. However, the expert's policy is often a function of a shared *latent structure* underlying those observable features that is invariant across settings. By leveraging data from multiple environments, we propose *Invariant Causal Imitation Learning* (ICIL), a novel technique in which we learn a feature representation that is invariant across domains, on the basis of which we learn an imitation policy that matches expert behavior. To cope with transition dynamics mismatch, ICIL learns a *shared* representation of causal features (for all training environments), that is independent from the *specific* representations of noise variables (for each of those environments). Moreover, to ensure that the learned policy matches the observation distribution of the expert's policy, ICIL estimates the energy of the expert's observations and uses a regularization term that minimizes the imitator policy's next state energy. Experimentally, we compare our methods against several benchmarks in control and healthcare tasks and show its effectiveness in learning imitation policies capable of generalizing to unseen environments.

## 1 Introduction

Strictly batch imitation learning aims to learn a policy that directly mimics the behaviour of experts, for which we only have access to a set of demonstrations: logged trajectories of observations and actions following the expert's policy [1–3]. We cannot interact online with the environment, let alone query the expert any further, nor do we have reward signals for supervision. This setting is relevant in real-world scenarios where live experimentation is risky or costly—such as healthcare and education.

Our aim is to learn an imitation policy in the strictly batch setting that faithfully matches the expert behaviour, while at the same time is able to generalize to unseen environments. In healthcare, learning a generalizable behaviour policy that could achieve expert performance in new environments is an important goal: As a means of providing clinical decision support, it could serve as an "individualized"

---

[*]Equal contribution.

35th Conference on Neural Information Processing Systems (NeurIPS 2021).

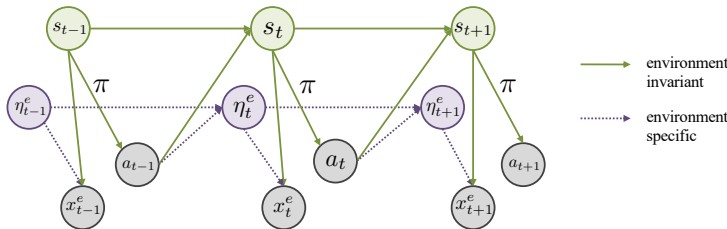

Figure 1: Causal diagram for the structure of environments. Expert demonstrations contain information about observations $x_t$ and actions $a_t$. We assume that observations are decomposable into (1) state representations $s_t$ that consist of the causal parents of the actions, and (2) noise representations $\eta_t$ that encapsulate any spurious correlations with the actions. To allow for dynamics mismatch, the transitions between the noise representations are specific to each environment. We want to recover the invariant state representation $s_t$ such that the learned policy $\pi(\cdot|s_t)$ generalizes well to new environments.

clinical guideline for actions that can be taken for different patients—especially in a hospital, region, or patient demographic from which we have no access to data during training. In this endeavor, a principal challenge is that the sets of expert demonstrations that we have access to may contain variables that induce selection bias, or are otherwise spuriously correlated with the expert's actions [4–7]. Directly learning an imitation policy from such data may lead to learning those spurious associations, thereby failing to generalize to unseen environments, and perpetuating any biases in the expert's behaviour.

However, in general it is likely that the expert's actions are only causally affected by a subset of the observed variables or by a shared latent structure [8, 9]. For instance, when imitating ideal driving behaviour, the background scenery might change, but the actions should only depend on car and road features. Another example includes the case when the lightning conditions in a room are changing, but physical dynamics of the environment are staying the same [7]. By leveraging expert trajectories from multiple different environments, our aim is to uncover this shared latent structure that causally determines expert actions, which allows us to eliminate the spurious associations and biases. In this way, the learnt policy will better be able to generalize to any unseen environments that share the same latent structure as those used for training.

As illustrated in Figure 1, we assume access to observations and actions from the expert's policy in the different environments $e$. The observations are functions of noise factors $\eta^e$ (which may differ across environments) and shared latent state representations $s$ (which is invariant across environments)— that encapsulate the causal parents of the expert's actions. Note that the observed features for an environment may simply be the union of $\eta^e$ and $s$, but they may also be any non-linear transformation of them. We shall operate in the setting where there are no hidden confounders, i.e. that we observe all variables that are affecting the expert's actions (and the next states that result from these actions).

In addition to spurious correlations, another difficulty stems from learning to imitate sequential behavior in the strictly batch setting itself: While behaviour cloning [10] provides an intrinsically batch solution, it ignores important information contained in the expert's roll-out distribution, and the learned policy may drift from the support of the distribution of states visited by the expert [11, 12].

**Contributions:** In this paper, we introduce *Invariant Causal Imitation Learning* (ICIL), a novel method that learns a causal representation of the expert's actions—which is used to build a generalizable imitation policy that matches the expert's behaviour. ICIL operates in the strictly batch setting and does not assume access to data from the target environments. By leveraging expert demonstrations from multiple different training environments, ICIL learns an (shared) invariant causal representation as well as an (environment-specific) noise representation. This accommodates dynamics mismatch across environments, while allowing the imitation policy to be learned by conditioning on the invariant causal representations. First, to satisfy the causal relationships in Figure 1, ICIL learns dynamics preserving representations and ensures that the learnt causal and noise representations are marginally independent by minimizing their mutual information. Second, to encourage the learnt imitation policy to stay within the support of the distribution of states visited by the expert's policy, ICIL estimates the energy of the expert's observations and uses a regularization term that minimizes the imitator policy's next state energy. Third, we evaluate ICIL against benchmarks for batch imitation learning in control and healthcare environments. We also empirically investigate directly using ideas from invariant risk minimization [6] to augment the loss function of existing batch imitation learning methods, and benchmark against their ability to generalize across environments.

| | Method | Environment | Offline | Dynamics mismatch | Sensory-shift (hidden confounders) | State-distribution matching | No access to target trajectories | Temporal aspect |
|---|---|---|---|---|---|---|---|---|
| Imitation Learning | Pomerleau [10] | Model-free | Yes | No | No | No | N/A | No |
| | Ho & Ermon [15] | Model-free | No | No | No | Model rollouts | N/A | Yes |
| | Kostrikov et al. [2] | Model-free | Yes | No | No | Adversarial off-policy matching | N/A | Yes |
| | de Haan *et al.* [8] | Model-free | No | No | No | No | N/A | Yes |
| Generaliz. in IL | Lu *et al.* [16] | Model-free | No | Yes | No | Model rollouts | No | Yes |
| | Kim *et al.* [17] | Model-based | No | Yes | No | Model rollouts | No | Yes |
| | Etsami *et al.* [18] | Model-free | Yes | No | Yes | No | No | Yes |
| IRM | Arjovsky *et al.* [6] | Model-free | Yes | N/A | N/A | N/A | Yes | No |
| ICIL (Ours) | | Model-based | **Yes** | **Yes** | No | Energy-based | **Yes** | **Yes** |

Table 1: Comparison of our proposed method with related works. ICIL operates in the strictly batch setting, allows for dynamics mismatch, does not require access to target trajectories, and incentivizes the imitation policy to stay in the support of the expert's distribution via energy-based regularization.

## 2  Related Works

We tackle the problem of learning generalizable policies in an offline setting using ideas from causal inference. As such, our work straddles the intersection of research in (1) strictly batch imitation, (2) invariant representation learning, and—more broadly—(3) causality in sequential decision-making.

**Strictly Batch Imitation Learning**: The simplest approach to imitation learning in the batch setting is behaviour cloning [10] which uses standard supervised learning techniques to learn an imitation policy that minimizes the negative log-likelihood of the observed demonstrator actions. However, behaviour cloning suffers from distributional shift as the learnt imitation policy cannot recover if it reaches a state out-of the distribution of the expert demonstrations [11–14]. To overcome this problem, [1, 14] propose incorporate dynamics-awareness by adding regularization to behaviour cloning by using norm-based penalties on the sparsity of implied rewards. Alternatively, [2] uses a distribution matching approach and propose an offline objective for estimating the distribution ratio of the imitator policy and the expert policy, while [3] jointly learn a policy function together with an energy-based model of the state distribution. However, none of the existing approaches consider the problem of *generalization* across environments and learning policies robust to spurious correlations.

**Invariant Risk Minimization**: In the supervised learning setting, Invariant Risk Minimization (IRM) [6] leverages data from multiple domains to learn a data representation that elicits an invariant predictor across the different environments. The training data from each environment corresponds to different interventions on the data generating process. Given data from several training environments, the IRM objective aims to find a representation such that there exists a classifier that is optimal across all training domains, i.e. that minimizes the empirical risk in each domain. This represents a challenging, bi-level optimization, and [6] propose the IRM-v1 objective which is a practical version to optimize. Through this optimization, the IRM objective should learn a predictor that only uses the causal parents of the target variable and that is thus invariant across environments. However, directly using IRM for our *sequential* problem setting is not desirable, since it does not take into account the effect of each action on the subsequent states. Nonetheless, we empirically investigate augmenting existing methods for batch imitation to use the IRM-v1 objective in conjunction with their defined imitation risk, and verify whether they are able to generalize across environments. In our experiments we observe that, in general, directly applying the IRM objective in this manner is not good enough.

**Generalization in Imitation Learning:** The problem of domain adaptation and transfer learning for the imitation learning setting has been tackled by several works so far. However, while they consider problems of dynamics-, embodiment-, and/or viewpoint-mismatch between the imitator and expert, existing methods assume access to demonstrations from the target environment [17, 19, 20], assume access to online interaction or simulators in the different environments [16], or focus on the different problem of hidden confounding [18, 21]. Another line of work that is related is learning from demonstrations and meta-learning. While works in meta-learning also aims to generalize learnt policies to new-tasks, they require access to one or more expert trajectories from the new task [22–27].

**Causality in Imitation Learning and Reinforcement Learning:** Several ideas from causality have been used to improve imitation learning and generalization in reinforcement learning. The idea of conditioning the imitation policy on the causal parents has been employed by [8] to avoid the problem of 'causal confusion' when learning a policy for the single environment setting. However, [8] requires

queering the expert or being able to perform interventions in the environment this is not possible in the batch setting. Similarly, [9, 28] also learn causal relationships between the observations, actions and rewards by performing/simulating the effect of interventions in the environment. Alternatively, [29] use ideas from Invariant Risk Minimization [6, 30] to learn optimal reinforcement learning policies that generalize across domains. Perhaps the most similar setting to ours is the one in [7] which studies the problem of generalization in reinforcement learning and also learn a representation that is shared across the domains. However, unlike our imitation setting, they assume access to a *known* reward signal, and focus on learning the causal ancestors of that reward to improve reinforcement learning [31].

To the best of our knowledge, we are the first to tackle the problem of learning generalizable imitation policies in the strictly batch setting. Table 1 summarizes main differences with relevant related works.

## 3 Problem Formalism

### 3.1 Imitation Learning

We work in the standard Markov decision process (MDP) setting: Let an environment be given by $e = (\mathcal{X}, \mathcal{A}, T, r, \gamma)$, with observations $x \in \mathcal{X}$, actions $a \in \mathcal{A}$, transition function $T \in \Delta(\mathcal{X})^{\mathcal{X} \times \mathcal{A}}$, reward function $r \in \mathbb{R}^{\mathcal{X} \times \mathcal{A}}$, and discount factor $\gamma$. Let $\pi \in \Delta(\mathcal{A})^{\mathcal{X}}$ be a policy with the induced occupancy measure $\rho_\pi(x, a) = (1 - \gamma) \sum_{t=0}^{\infty} \gamma^t p(x_t = x, a_t = a | x_t \sim T(\cdot | x_{t-1}, a_{t-1}), a_t \sim \pi(\cdot | x_t))$ of observations and actions, and let $\rho_\pi(x) = \sum_{a \in \mathcal{A}} \rho_\pi(x, a)$ be the observation occupancy measure.

Unlike in the reinforcement learning setting, where the aim is to learn a policy $\pi(\cdot | x)$ that maximizes the cumulative sum of some known reward signal, in imitation learning the reward is neither known nor observed. Instead, we only have access to a dataset of trajectories $\mathcal{D} = \{\tau_i\}_{i=1}^{N}$ from a demonstrator policy $\pi_D$, where each trajectory $\tau \sim \pi_D = (x_t, a_t, x_{t+1})_{t=0,\dots}$ consists of a sequence of observation, action, next observation tuples that are sampled as $a_t \sim \pi_D(\cdot | x_t)$ and $x_{t+1} \sim T(\cdot | x_t, a_t)$.

The goal of imitation learning is to seek an imitation policy $\pi$ that minimizes the following risk:

$$R(\pi) = \mathcal{L}(\pi, \pi_D) \tag{1}$$

where $\mathcal{L}$ is a choice of loss function. Now, if we were in the online setting, we would have access to the environment (or a simulator), with which we can interactively perform distribution matching by minimizing the divergence between the expert's state occupancy $\rho_D$ and the imitator's state occupancy $\rho_\pi$ [15, 32–34]. One example is to use the (forward) KL divergence: $\mathcal{L}(\pi, \pi_D) = D_{KL}(\rho_D || \rho_\pi)$ [34]. However, in the offline setting we have no further access to the environment. As noted above, the simplest solution is behaviour cloning (BC) [10, 35, 36], which minimizes the negative log-likelihood of the demonstrator's actions. However, by disregarding the distribution of the expert's observations, imitation policies learnt by BC often result in compounding error when deployed in practice [11–14].

### 3.2 Imitation Learning from Multiple Environments

Consider a *family* of environments $\mathcal{M} = \{(\mathcal{X}^e, \mathcal{A}, T^e, r^e, \gamma) \mid e \in \mathcal{E}\}$ with observations $x^e \in \mathcal{X}^e$, actions $a \in \mathcal{A}$, transition function $T^e \in \Delta(\mathcal{X})^{\mathcal{X} \times \mathcal{A}}$, reward function $r^e \in \mathbb{R}^{\mathcal{X} \times \mathcal{A}}$, and discount factor $\gamma$. This is the primary setting that we shall operate in. Note that the action space and discount factor do not change between environments. For notational simplicity, when considering the union over environments, we shall drop the index $e$. We assume offline access to a dataset of recorded trajectories from the expert policy $\pi_D$ in a set of training environments $\mathcal{E}_{train} \subset \mathcal{E}$, $\mathcal{D} = \{\{(\tau_i^e)_{i=1}^{N_e} \mid e \in \mathcal{E}_{train}\}$. Each trajectory $\tau^e \sim \pi_D = (x_t^e, a_t, x_{t+1}^e)_{t=0,\dots}$ consists of a sequence of environment specific observations, expert actions and next observations sampled as $a_t \sim \pi_D(\cdot | x_t^e)$ and $x_{t+1}^e \sim T^e(\cdot | x_t^e, a_t)$.

In the presence of multiple environments, our goal is to learn a policy $\pi \in \Delta(\mathcal{A})^{\mathcal{X}}$ that matches the expert behaviour in all possible environments $\mathcal{E}$ that share a certain structure for the observations and the transition dynamics. In particular, this involves finding a policy that *generalizes* well across these related environments $e \in \mathcal{E}$—that is, the policy should ideally minimize the imitation risk across them:

$$\max_{e \in \mathcal{E}} R^e(\pi) = \mathcal{L}^e(\pi, \pi_D) \tag{2}$$

where each $\mathcal{L}^e$ explicitly depends on the characteristics of the environment $e$. Note that since we know nothing specific about $\mathcal{E}$, it is difficult to optimize for this directly. That said, if we make mild assumptions about the "relatedness" of these environments, we can learn policies that generalize well.

**Structure of Observations:** First, we assume there is a shared latent structure underlying the observations from different environments—on which the expert policy depends. Finding such a structure would let us discard irrelevant factors as inputs to the learnt policy, improving generalization [6, 7, 37]:

**Assumption 3.1.** *(Shared Latent Structure) Consider decomposing the observations $x^e \in \mathcal{X}^e$ in each environment $e \in \mathcal{E}$ into two components: an invariant representation $s \in \mathcal{S}$ and noise terms $\eta^e \in \mathcal{Z}^e$ (i.e. spurious correlations), such that $x^e = q(s, \eta^e)$ for some invertible transformation $q : \mathcal{S} \times \mathcal{Z}^e \to \mathcal{X}^e$. There exists some $q$ such that $\pi_D$ only depends on $s$, and space $\mathcal{S}$ is non-empty.*

In other words, we assume that the demonstrator's policy $\pi_D$ depends only on information that is shared across the environments, i.e. the state variables $s$ are the causal parents of the expert action $a \sim \pi_D(\cdot \mid s)$. As illustrated in Figure 1, the state variables and the noise terms are responsible for generating the patient observations, but the policy depends only on the state variables. Thus, we allow different environments to have different $p(x)$ marginals (as well as different $p(a|x)$). This allows environments to have different structure. The only requirement is that the environments are the same as far as the task is concerned. This means that there exists some $\mathcal{S}$ such that $p(s)$ marginals should be the same (as well as $p(a|s)$). Learning such a representation that is invariant satisfies the standard Environment Invariance Constraint [38]. While this set-up is similar to the one in [7], a crucial difference is that we have no access to any reward functions whatsoever, and that we must learn an imitation policy in a strictly batch setting.

Note that the latent structure induced by the state variables is *shared* across the different environments. This means that the transition dynamics for the state representation $p(s_{t+1} \mid s_t, a_t)$ remain invariant across the environments. On the other hand, as different environments may be characterized by different types of spurious correlations, to allow for flexibility in their structure and evolution, we consider that the transition dynamics of the noise terms $p^e(\eta^e_{t+1} \mid \eta^e_t, a_t)$ are *specific* to each environment. Our goal, then, is to learn a **generalizable policy** $\pi$—that is, one that depends only on $s$.

**Structure of Environments:** Second, to learn a policy that depends only on $s$, we must assume that the available training environments are actually different, so that we can learn the invariant state representation using the data from these environments and separate it from the noise representation:

**Assumption 3.2.** *(Environment Interventions) Each available training environment $e$ corresponds to a hard [39] or soft [40] intervention on one or more dimensions of that environment's observation space (where these dimensions do not constitute any causal parents of the demonstrator's actions).*

To ensure that a generalizable policy actually exists, Assumption 3.1 requires that $\mathcal{S}$ be non-empty across all environments. Here, to ensure that the space $\mathcal{S}$ can actually be learned, Assumption 3.2 requires that $\mathcal{Z}$ be non-empty across the training environments. Note that we require that the interventions inducing the different environments *not* be on the causal parents of the action, such that Assumption 3.1 is not violated.

Overall, in our setting $p(x)$ and $p(a \mid x)$ can differ between the multiple environments. However, Assumption 3.1 enforces that the environments and tasks are the same modulo noise, i.e. that there exists some non-empty $\mathcal{S}$ such that $p(s)$ and $p(a \mid s)$ are the same between them. In other words, we have a set of environments that are different (i.e. the dynamics of $x_t$ are different), but the task being performed by the agent is the same (i.e. the dynamics of $s_t$ are the same). This setting applies to the case when lightning conditions in a room are changing, but physical dynamics of the environment are staying the same [7] or when weather conditions are changing, but driving behaviour and dynamics are staying the same. We provide additional explanations and definitions of environment interventions in Appendix A.

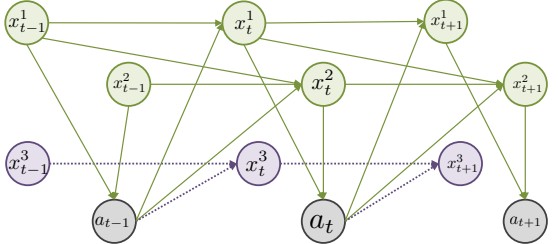

Figure 2 shows a simple example where each observation $x_t$ represents a union of the causal parents of the action (state variables) $s_t = \{x^1_t, x^2_t\}$ and the spurious correlations (noise variables) $\eta_t = \{x^3_t\}$. To satisfy Assumption 3.2, the different environments need to correspond to interventions on $x^3_t$. And to satisfy Assumption 3.1,

Figure 2: Causal diagram illustrating temporal dependencies between causal parents of action $\{x^1_t, x^2_t\}$ and noise variables $\{x^3_t\}$. Different environments are induced by different interventions on the noise variables.

$x_t^1$ and $x_t^2$ must not be intervened on. Our aim is to find a representation $s$ of the causal parents $\{x_t^1, x_t^2\}$ of the actions, as well as the mapping between them and the actions $a_t$, that mimics the expert's policy.

Finally, similarly to [7], we also assume that the observations $x_t$ at timestep $t$ can only affect the actions $a_t$ and the observations at the next timestep $t + 1$:

**Assumption 3.3.** *(Temporal Causal Mechanism) Let $x^i$ and $x^j$ be any two components of the observation $x$ at timestep $t$. Then:*

$$x_{t+1}^i \perp\!\!\!\perp x_{t+1}^j \mid x_t, a_t \tag{3}$$

Note that Assumption 3.3 simply serves to place us within the standard MDP setting: It ensures Markovianity of the temporal transitions, that only the observations $x_t$ at time $t$ will contain the causal parents of the action $a_t$, and that $x_t$ and $a_t$ are the only factors that determine the next observation $x_{t+1}$.

## 4 Invariant Causal Imitation Learning for Domain Generalization

The goal of our Invariant Causal Imitation Learning (ICIL) algorithm is to learn a representation of the state variables $s$ that is invariant across domains, and an imitation policy $\pi$ that depends on this causal representation and matches the demonstrator's behaviour. We operate in the strictly batch setting, and our aim is for $\pi$ to generalize to unseen environments $e \in \mathcal{E}$ given the above structural assumptions.

### 4.1 Learning Invariant Causal Representations

To achieve our goal, we decompose the observations $x_t^e$ in each environment $e$ into a representation $s_t = \phi(x_t^e; \theta_s)$ for the causal features of the action $a_t$, and another representation $\eta_t^e = \mu^e(x_t^e; \theta_\eta^e)$ for the noise variables, where $\theta_s$ and $\theta_\eta^e$ are the learnable parameters of $\phi$ and $\eta$. Since the causal parents of the action are invariant across the environments, the state representation model $\phi : \mathcal{X} \to \mathcal{S}$ is the same across all environments. On the other hand, $\mu^e : \mathcal{X} \to \mathcal{Z}^e$ is environment-specific in order to allow for dynamics mismatch of the noise variables between the different environments.

In order to satisfy the causal diagram in Figure 1 and to learn a minimal causal representation, we need the following conditions to be satisfied: (1) $s_t$ should be *invariant* across the environments, (2) $s_t$ and $\eta_t^e$ should be *dynamics-preserving*, and (3) $s_t$ and $\eta_t^e$ should be *independent* from each other.

Firstly, to fulfill condition (1) we train an environment classifier on the shared state representation $c_s : \mathcal{S} \to |\mathcal{E}_{train}|$, parameterized by $\theta_c$ using the cross-entropy loss. Similarly to [7], in order to build a state representation that is invariant across domains, we use an adversarial loss [41] that maximizes the entropy of the classifier: $H(c_s(\phi(x_t; \theta_s); \theta_c))$. This gives us the following practical loss function:

$$\mathcal{L}_{inv}(\theta_s) = \sum_{e \in \mathcal{E}_{train}} \mathbb{E}_{x_t^e \sim \rho_D^e} - H(c_s(\phi(x_t^e; \theta_s); \theta_c)) \tag{4}$$

Out of all possible representations that are invariant, we specifically seek one that also preserves the transition dynamics, fulfilling condition (2). To ensure that the state and noise representations are dynamics-preserving, we also learn the transition dynamics for the state variables $g_s : \mathcal{S} \times \mathcal{A} :\to \mathcal{S}$, such that $\hat{s}_{t+1} = g_s(s_t, a_t; \theta_{g_s})$ and the environment specific transition dynamics for the noise variables: $g_\eta^e : \mathcal{Z}^e \times \mathcal{A} \to \mathcal{Z}^e$, such that $\hat{\eta}_{t+1}^e = g_\eta^e(\eta_t^e, a_t; \theta_{g_\eta}^e)$. To reconstruct $x_{t+1}$ we also learn $\psi : \mathcal{S} \times \mathcal{Z}^e \to \mathcal{X}$ such that $\hat{x}_{t+1}^e = \psi(s_{t+1}, \eta_{t+1}^e; \theta_\psi)$. This yields the following practical loss function:

$$\mathcal{L}_{dyn}(\theta_s, \theta_{g_s}, \{\theta_\eta^e, \theta_{g_\eta}^e\}_{e \in \mathcal{E}_{train}}, \theta_\psi) = \sum_{e \in \mathcal{E}_{train}} \mathbb{E}_{x_{t+1}^e, a_t, x_{t+1}^e \sim \rho_D^e} \|x_{t+1}^e - \hat{x}_{t+1}^e\|^2 \tag{5}$$

Note that while an alternative approach could consider directly building an invertible mapping from $x^e$ to $(s, \eta^e)$, the motivation for decoding $s_{t+1}$ and $\eta_{t+1}^e$ into $\hat{x}_{t+1}^e$ is twofold. In addition to learning dynamics-preserving representations, as we will see in Section 4.2, this also allows us to compute the energy of the next state obtained by following the imitation policy and enforcing this to be similar to the distribution of states visited by the expert's policy.

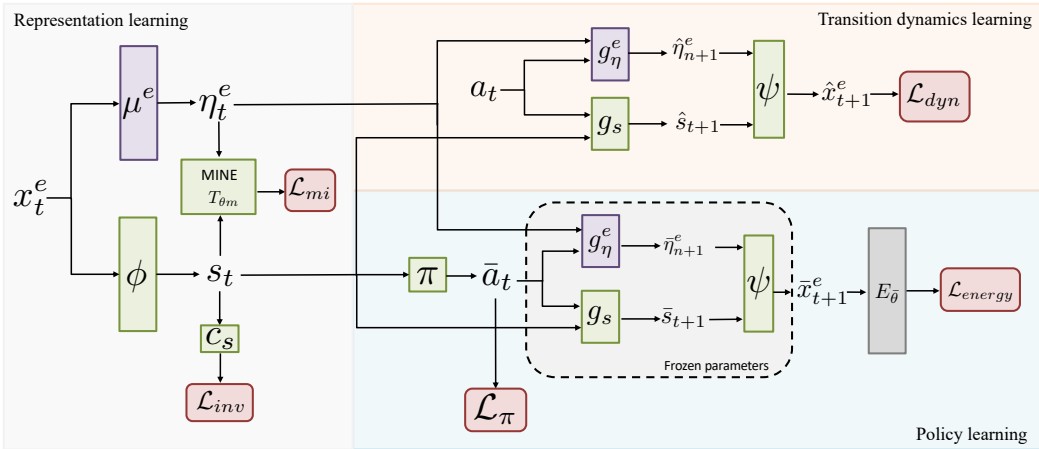

Figure 3: Block diagram of our model. ICIL decomposes the observations $x_t^e$ into an invariant causal representation $s_t$ and an environment specific noise representation $\eta^e$. To obtain an invariant representation, we maximize entropy of an environment classifier that receives as input $s_t$ ($\mathcal{L}_{inv}$). Moreover, the state and noise representations are learnt to be dynamics preserving by minimizing the prediction error of the next observation ($\mathcal{L}_{dyn}$) and independent by minimizing their mutual information ($\mathcal{L}_{mi}$). We learn a generalizable imitation policy that is conditioned on the invariant causal representation ($\mathcal{L}_{\pi}$) and to ensure that the learnt policy matches the distribution of the expert's observations, we minimize the imitator's policy next state energy ($\mathcal{L}_{energy}$).

Finally, to ensure that the state representation and the noise representation are marginally independent per condition (3), we minimize the mutual information between them. We use the Mutual Information Neural Estimation (MINE) framework [42], which provides a way for estimating the mutual information using neural networks. In particular, MINE uses a neural information measure $I(U, V)$ to approximate the mutual information between random variables $U$ and $V$. Let $T_{\theta_m}$ be a statistics network parametrized by $\theta_m$. MINE estimates $I(U, V)$ by ascending the gradient of the following:

$$I(U, V) = \sup_{\theta_m} \mathbb{E}_{\mathbb{P}_{UV}^{(n)}}[T_{\theta_m}] - \log(\mathbb{E}_{\mathbb{P}_U^{(n)} \otimes \mathbb{P}_V^{(n)}}[e^{T_{\theta_m}}]) = \sup_{\theta_m} I(U, V; \theta_m) \tag{6}$$

where $\mathbb{P}_{UV}$ is the joint measure of $(U, V)$ and $\mathbb{P}_U = \int_{\mathcal{V}} dP_{UV}$, $\mathbb{P}_V = \int_{\mathcal{U}} dP_{UV}$ are the marginal distributions. $\mathbb{P}^{(n)}$ denotes the empirical distribution associated with $n$ i.i.d samples. As noted in [42], the neural information measure $I(U, V)$ can approximate the mutual information with arbitrary accuracy. We therefore add the following practical loss function to our optimization objective, which seeks to minimize the mutual information between the state representation and noise representation:

$$\mathcal{L}_{mi}(\theta_s, \{\theta_\eta^e\}_{e \in \mathcal{E}_{train}}) = \sum_{e \in \mathcal{E}_{train}} \mathbb{E}_{x_t^e \sim \rho_D^e} I(\phi(x_t^e; \theta_s), \mu(x_t^e; \theta_\eta^e); \theta_m) \tag{7}$$

Note that the parameters $\theta_m$ of the statistics network $T_{\theta_m}$ used for computing the mutual information are updated through gradient ascent on $I(U, V; \theta_m)$.

## 4.2 Matching Expert Behaviour in a Strictly Batch Setting

On the basis of the causal representation $s$, we shall learn a generalizable policy $\pi$ (parameterized by $\theta_\pi$) in the strictly batch setting, such that it matches the demonstrator's behaviour. To begin, we first condition $\pi$ on the representation $s_t$ and minimize the negative log-likelihood of the expert's actions:

$$\mathcal{L}_{\pi}(\theta_\pi, \theta_s) = \sum_{e \in \mathcal{E}_{train}} -\mathbb{E}_{x_t^e, a_t \sim \rho_D^e} \log \pi(a_t \mid \phi(x_t^e; \theta_s); \theta_\pi) \tag{8}$$

However, having only this objective corresponds to performing behaviour cloning, which has well-known limitations [11–14]. To mitigate compounding error, we want some form of added regularization to incentivize the imitation policy to stay within the distribution of the expert's observations.

In the online setting, a popular approach is to make sure that the rollout distribution of the imitating policy matches the rollout distribution of the expert's policy—for instance, by minimizing some form

of divergence between their induced occupancy measures. However, this requires interactive access to the real environment or simulator to perform rollouts of intermediate policies—which is not possible in our setting. Instead, we propose a method that takes advantage of the learnt transition dynamics. For any current observation $x_t \sim \rho_D$, we shall encourage the next observation $\bar{x}_{t+1}$ obtained by following the imitation policy $\bar{a}_t \sim \pi(\cdot \mid x_t)$ to remain within the occupancy measure of the expert.

Consider approximating the expert's occupancy measure using an Energy Based Model (EBM) such that $\rho_D(x) = \frac{\exp(-E_{\bar{\theta}}(x))}{Z(\bar{\theta})}$ where the function $E_{\bar{\theta}}(x) : \mathcal{X} \rightarrow \mathbb{R}$ is the energy function and $Z(\bar{\theta}) = \int_x -E_{\bar{\theta}}(x)dx$ is the partition function. We parameterize $E_{\bar{\theta}}$ by a neural network. It is not possible to train the EBM directly through maximum likelihood because $Z(\bar{\theta})$ involves integrating over the entire input domain of $x$ which is impractical. Instead, we use contrastive divergence to pre-train the energy function $E_{\bar{\theta}}$ [43, 44]. Contrastive divergence lowers the energy of the observations coming from the expert's occupancy distribution and increases the energy of the observations outside of the expert's occupancy distribution. Refer to Appendix B for details on how we train the EBM.

To incentive the imitation policy to stay within the distribution of the expert's observations, we train it to minimize the energy of the next observation obtained by following $\pi$ given the current observation:

$$\mathcal{L}_{energy}(\theta_\pi; \theta_s, \theta_{g_s}, \{\theta_\eta^e, \theta_{g_\eta}^e\}_{e \in \mathcal{E}_{train}}, \theta_\psi) = \sum_{e \in \mathcal{E}_{train}} \mathbb{E}_{\substack{x_t^e \sim \rho_D^e \\ s_t = \phi(x_t^e), \eta_t^e = \mu^e(x_t^e) \\ \bar{a}_t \sim \pi(\cdot|s_t) \\ \bar{x}_{t+1} = \psi(g_s(s_t, \bar{a}_t), g_\eta^e(\eta_t^e, \bar{a}_t))}} E_{\bar{\theta}}(\bar{x}_{t+1}^e) \quad (9)$$

This effectively assigns a high "reward" to the imitation policy for staying within high-density areas of the expert's occupancy measure, and a low "reward" for straying from it. This can be seen as an adaptation of online imitation methods [45, 46] where the expectation would be instead over $x_t \sim \rho_\pi$.

We illustrate in Figure 3 all of the components of the our ICIL model. Further details and the full algorithm for optimizing ICIL can be found in Appendix C.

## 5 Experiments

We perform experiments on OpenAI gym tasks [47] and on an ICU dataset from the MIMIC III database [48]. In both cases, we generate data from multiple domains by augmenting the feature space with noise variables (spurious correlations).

**Benchmarks** We compare ICIL[1] against standard methods for strictly batch imitation learning: Behaviour Cloning (BC) [10]; Reward-regularized Classification for Apprenticeship Learning (RCAL), which incorporates dynamics-awareness through a sparsity regularization on the implied rewards [14]; ValueDICE (VDICE) [2], which uses an off-policy objective to estimate distribution ratios needed for distribution matching; as well as Energy-based Distribution Matching (EDM) [3], which jointly learns the imitator policy with an energy model of its state distributions. These methods seek to find a policy that approximately matches the expert's behaviour from a single environment, and were not designed with generalization in mind. Hence we augment these benchmark by using the IRMv1 objective [6] in conjunction with their originally defined imitation risk to obtain the additional benchmarks: BC-IRM, RCAL-IRM, VDICE-IRM, and EDM-IRM. More details about how we used the invariance-based penalty from IRM [6] to augment these existing methods such that they may learn generalizable policies can be found in Appendix D. Implementation details about all benchmarks and the hyperparameter settings used can be found in Appendix F.3.

### 5.1 Evaluation on OpenAI Gym

We perform experiments on the following control tasks from OpenAI gym [47]: Acrobot [49], Cartpole [50], LunarLander [47] and BeamRider [51]. For each task, we use pre-trained RL agents from RL Baselines Zoo [52] and Stable OpenAI Baselines [53] to obtain expert policies. We then follow an approach similar to the one in [7] to obtain datasets with demonstrations from the expert in two different environments. In particular, for Acrobot [49], Cartpole [50] and LunarLander [47] we add spurious correlations to the state space of each control task and an environment identifier. The

---

[1]The code for ICIL can be found at https://github.com/vanderschaarlab/mlforhealthlabpub and at https://github.com/ioanabica/Invariant-Causal-Imitation-Learning.

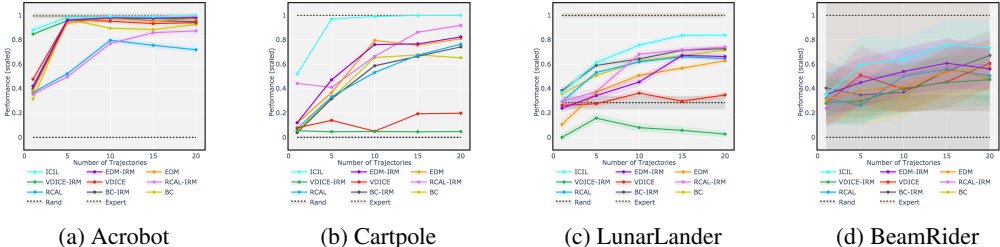

| (a) Acrobot | (b) Cartpole | (c) LunarLander | (d) BeamRider |

Figure 4: Evaluation on OpenAI gym environments. $x$-axis indicates the number of trajectories (in $\{1, 5, 10, 15, 20\}$) with expert demonstrations from each training environment given as input to each benchmark and $y$-axis represents the average return of the learnt imitation policy on the test environments, scaled between 1 (expert performance) and 0 (random policy performance).

spurious correlations in each environment are different multiplicative factors of a subset of variables in the original state space. The invariant causal state is represented by the original variables in the state space of each control task. We train the benchmarks on demonstrations from two environments with $1\times$ and $2\times$ multiplicative factors for the spurious correlations and we test on an environments with multiplicative factors sampled from $\mathcal{U}(-1, 1)$. For BeamRider [51], similarly to [7], different camera angles are used for the training and testing environments. In particular, we use two training environments where the game frames are rotated by 10 degrees to the left and to the right respectively, while the test environment has no rotation. The rotation is applied to the entire frame for all trajectories in each environment. However, note that, despite the rotation, the dynamics for the state variables and how they influence the action stay the same. Further details about the train and test environments can be found in Appendix F.3.

We vary the number of demonstrated trajectories from each environment that we give as input to each benchmark and we evaluate them on the average return obtained by deploying the learnt imitation policies on the test environment. Figure 4 shows the mean results and standard errors obtained across 10 runs where for each run we train the benchmarks on different trajectories from the train environments and we evaluate on a test environment with newly sampled multiplicative factors for computing the spurious correlations. We notice that our method consistently outperforms the benchmarks and is capable of generalizing better to the unseen target environments. Moreover, we generally found that using the IRMv1 objective [6] together with existing methods for strictly batch imitation learning did not improve performance and resulted in more unstable training. For additional results, see Appendix F where we perform ablation studies to investigate the impact of the different terms in the loss function used to train ICIL on overall performance, compare performance on train vs. test environments and also evaluate robustness to increasing the size of the spurious correlations.

## 5.2  Evaluation on MIMIC-III

We also perform experiments on a healthcare dataset with Intensive Care Unit (ICU) patients extracted from the Medical Information Mart for Intensive Care (MIMIC-III) database [48]. The dataset consists of trajectories of clinical measurements (e.g. heart rate, respiratory rate) recorded every hour. The aim is to learn a generalizable policy for the action of putting patients on the mechanical ventilator.

We define three environments, two for training and one for testing, each consisting of 2000 independent patient trajectories from MIMIC-III. We augment the original feature space by adding spurious correlations (noise variables) that are the same as the expert actions with probabilities $p = 0.1$ and $p = 0.2$ in the training environments and with probability $p = 0.8$ in the testing environment.

In a real setting, such spurious correlations are commonplace. For instance, consider some hospitals (i.e. training environments) where selection bias is present, such that patients with a certain otherwise irrelevant comorbidity happen to receive a treatment more often [54–57]. However, learning an imitation policy that takes into account such a comorbidity when assigning the patient's treatment would fail to generalize to hospitals where fewer patients suffer from this comorbidity but should still receive the treatments. More details about the dataset can be found in Appendix F.3.

|  | Mechanical ventilator | | |
| Benchmark | ACC | AUC | APR |
|---|---|---|---|
| BC | $0.783 \pm 0.001$ | $0.762 \pm 0.002$ | $0.692 \pm 0.001$ |
| RCAL | $0.790 \pm 0.002$ | $0.771 \pm 0.002$ | $0.697 \pm 0.002$ |
| VDICE | $0.794 \pm 0.001$ | $0.784 \pm 0.001$ | $0.716 \pm 0.001$ |
| EDM | $0.786 \pm 0.003$ | $0.741 \pm 0.011$ | $0.682 \pm 0.005$ |
| BC-IRM | $0.791 \pm 0.002$ | $0.767 \pm 0.003$ | $0.696 \pm 0.002$ |
| RCAL-IRM | $0.789 \pm 0.002$ | $0.766 \pm 0.003$ | $0.694 \pm 0.003$ |
| VDICE-IRM | $0.766 \pm 0.001$ | $0.730 \pm 0.001$ | $0.694 \pm 0.001$ |
| EDM-IRM | $0.781 \pm 0.004$ | $0.717 \pm 0.015$ | $0.673 \pm 0.007$ |
| ICIL | $\mathbf{0.855 \pm 0.003}$ | $\mathbf{0.856 \pm 0.004}$ | $\mathbf{0.789 \pm 0.004}$ |

Table 2: Evaluation on MIMIC-III in terms of action-matching. We compare the actions selected by the benchmark imitation policies with the ones from the clinical expert policy in the test environment and report the accuracy (ACC), the area under the receiving operator characteristic curve (AUC) and the area under the the precision-recall curve (APR).

Since MIMIC-III is an entirely offline dataset, it is not possible to compute average returns for running the policies in the test environment. Instead, we evaluate the benchmarks in terms of action matching on the test environment. We report in table 2 the mean accuracy (ACC), the mean area under the receiving operator characteristic curve (AUC), the mean area under the the precision-recall curve (APR) and their standard deviations over 10 runs. We notice that ICIL learns a policy that best discards the spurious correlations present in the training environment to learn a generalizable policy for putting patients on the mechanical ventilator that best matches the expert's actions on the test environment.

## 6 Discussion

In this paper, we tackle the problem of learning generalizable imitation policies in the strictly batch setting. Our ICIL model leverages ideas from causality and learns an invariant state representation that minimizes the presence of spurious correlations. By conditioning the imitation policy on this state representation, we obtain a policy that generalizes to environments with the same shared latent structure, but with different noise distribution and dynamics. ICIL also matches expert behaviour by incentivizing the learnt imitation policy to stay within the expert's observations distribution.

In terms of limitations, we believe that future work should consider providing theoretical insights and error bounds on the generalization error. In addition, to be able to learn an invariant state representation, our method requires demonstrated trajectories from at least two training environments with different interventions on the noise variables (spurious correlations), and the method cannot be used if such data is not available in practice. Finally, we bear in mind that—as with any other imitation learning method that aims to match the expert's policy—ICIL can have potential negative societal impacts if the expert's policy is flawed in the first place. Thus, in sensitive applications such as clinical decision support, care must be taken to prevent potentially negative feedback loops.

## Acknowledgments

We would like to thank the reviewers for their valuable feedback. The research presented in this paper was supported by The Alan Turing Institute, under the EPSRC grant EP/N510129/1, by Alzheimer's Research UK (ARUK), by the US Office of Naval Research (ONR), and by the National Science Foundation (NSF) under grant number 1722516.

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
