# A Structure of environment and interventions

Similarly to [6], we consider that all environments have the same underlying Structural Causal Model (SCM) and that the different environments correspond to different interventions on the SCM. We provide here the formal definition for SCMs and interventions.

**Definition A.1.** *(Structural Causal Model) [6]: A structural causal model (SCM) $\mathcal{C} = (S, N)$ governing the random vector $X = (X_1, \ldots X_m)$ is a collection $S$ of $m$ assignments:*

$$S_j : X_j \leftarrow f_j(Pa(X_j), N_j), \text{ for } j = 1, \ldots m \tag{10}$$

*where $Pa(X_j) \subseteq \{X_1, X_2, \ldots X_m\}/\{X_j\}$ are the parents of $X_j$ and the $N_j$ are the independent noise variables. We say that $X_i$ causes $X_j$ if $X_i \in Pa(X_j)$.*

**Definition A.2.** *(Intervention) [6]: Consider a SCM $\mathcal{C} = (S, N)$. An intervention $e$ on $\mathcal{C}$ consists of replacing one or several of its structural equations to obtain an intervened SCM $\mathcal{C}^e = (S^e, N^e)$ with structural equations:*

$$S_j^e : X_j^e \leftarrow f_j(Pa(X_j^e), N_j^e), \text{ for } j = 1, \ldots m \tag{11}$$

*The variable $X^e$ is intervened on if $S_i \neq S_i^e$ or $N_i \neq N_i^e$.*

In our setting, the variables forming the SCM are the different observations and actions at each timestep. Moreover, Assumption 3.3 that ensures the Markovianity of the temporal transitions restrict the relationships that can be present in the SCM. In addition, Assumption 3.2 requires that the interventions to not be on the causal parents of the action.

Soft interventions [40] do not remove any edges in the causal graph induced by the SCM, but instead modify the conditional probability distributions of the variables intervened on. On the other hand, hard interventions [39] on a variable remove all incoming edges from the parents of that variables.

# B   Train energy based model

We train the energy-based model for the expert demonstrations using Persistent Contrastive Divergence [44]. To sample from the energy-based model, we use Markov Chain Monte Carlo using Langevin Dynamics [58]. Algorithm 1 outlines the method used to learn the energy-based model $E_{\bar{\theta}}$ of the expert observations. See [44] for details on training EBMs in this manner.

---

**Algorithm 1** Learning energy-based model $E_{\bar{\theta}}$ of expert demonstrations

---

1: **Input:** Dataset with expert demonstrations $\mathcal{D}$, number of steps $K$, step size $\alpha$, noise variance $\sigma$, Mini-batch size $N$
2: **Initialize:** energy-based model parameters $\bar{\theta}$, buffer $\mathcal{B} \leftarrow \varnothing$
3: **while** not converged **do**
4:       Sample $N$ positive samples from expert demonstrations $x_i^+ \sim \mathcal{D}$
5:       Sample initial negative samples: $x_i^0 \sim \mathcal{B}$ with 95% probability and $x_i^0 \sim \mathcal{U}(-1, 1)$ otherwise
6:       **for** sample step $k = 1$ to $K$ **do**                    ▷ Generate sample via Langevin dynamics
7:             $\tilde{x}_i^k \leftarrow \tilde{x}_i^{k-1} - \alpha \cdot \nabla_x E_{\bar{\theta}}(\tilde{x}_i^{k-1}) + \omega$, where $\omega \sim \mathcal{N}(0, \sigma), \forall i \in \{1, \ldots N\}$
8:       **end for**
9:       $x_i^- = \Omega(x_i^K), \forall i \in \{1, \ldots N\}$                    ▷ $\Omega$: stop gradient operator
10:       Contrastive divergence loss $\mathcal{L}_{CD} = \frac{1}{N}\sum_i E_{\bar{\theta}}(x_i^+) - E_{\bar{\theta}}(x_i^-)$
11:       Regularization loss: $\mathcal{L}_{RG} = \frac{1}{N}\sum_i E_{\bar{\theta}}(x_i^+)^2 + E_{\bar{\theta}}(x_i^-)^2$
12:       Update parameters $\bar{\theta}$ by backpropagating $\nabla_{\bar{\theta}}(\mathcal{L}_{CD} + \mathcal{L}_{RG})$
13:       Add samples to buffer: $\mathcal{B} \leftarrow \mathcal{B} \cup \{x_i^-\}_{i=1}^N$
14: **end while**

---

For the Acrobot [49], CartPole [50], LunarLander [47] control tasks and the experiments on the healthcare dataset extracted from MIMIC III [48], we use a neural network with 2 fully-connected hidden layers of size 64 and with ReLU activation to define $E_{\bar{\theta}}$. For Acrobot [49], CartPole [50], LunarLander [47], we set the hyperparameters to number of steps $K = 100$, step size $\alpha = 0.01$, noise variance $\sigma = 0.01$ and mini-batch size $N = 64$. We optimize $\bar{\theta}$ by using the Adam optimizer for 1000 training iterations with the learning rate set to 0.001. For the experiments on the healthcare dataset extracted from MIMIC III [48], we use the following hyperparameters for the number of steps $K = 50$, step size $\alpha = 0.01$, noise variance $\sigma = 0.01$ and mini-batch size $N = 128$. We optimize $\bar{\theta}$ by using the Adam optimizer for 1000 training iterations with the learning rate set to 0.0005.

To define $E_{\bar{\theta}}$ for the BeamRider Atari environment [51] we use a convolutional neural network with 3 convolutional layers with 32-64-64 filters, followed by a fully connected layer of size 64, with all layers followed by ReLU activations. The hyperparameters are set as follows: number of steps $K = 100$, step size $\alpha = 0.01$, noise variance $\sigma = 0.01$ and mini-batch size $N = 64$. $\bar{\theta}$ is optimized by using the Adam optimizer for 1000 training iterations with the learning rate set to 0.001.

# C Algorithm

Algorithm 2 provides the pseudo-code for training Invariant Causal Imitation Learning (ICIL).

---

**Algorithm 2** Invariant Causal Imitation Learning

---

1: Input: Dataset with expert demonstrations $\mathcal{D}$, learning rate $\lambda$, mini-batch size $N$
2: Initialize: $\theta_s, \theta_{g_s}, \{\theta_\eta^e, \theta_{g_\eta}^e\}_{e \in \mathcal{E}_{train}}, \theta_\psi, \theta_c, \theta_\pi, \theta_m$
3: **while** not converged **do**
4:     Sample mini-batch of N demonstrations $(x_i^{e_i}, a_i, x_{i+1}^{e_i}) \sim \mathcal{D}$
5:     Sample permutation of $[N] = \{1, \dots N\}$ from uniform distribution over the set of all permutations $S_N$: $\kappa \sim \mathcal{U}(S_N)$
6:     **ICIL update**:
7:     Invariance loss: $\mathcal{L}_{inv} = \frac{1}{N} \sum_{i=1}^{N} -H(c_s(\phi(x_i^{e_i})))$
8:     Transition dynamics loss:

$$\mathcal{L}_{dyn} = \frac{1}{N} \sum_{i=1}^{N} \|x_{i+1}^{e_i} - \psi(g_s(\phi(x_i^{e_i}), a_i), g_\eta^{e_i}(\mu^{e_i}(x_i^{e_i}), a_i))\|^2$$

9:     Mutual information loss:

$$\mathcal{L}_{mi} = \sum_{i=1}^{N} T_{\theta_m}(\phi(x_i^{e_i}), \mu(x_i^{e_i})) - \log(\sum_{i=1}^{N} \exp T_{\theta_m}(\phi(x_i^{e_i}), \mu(x_{\kappa(i)}^{e_{\kappa(i)}})))$$

10:     Policy loss:

$$\mathcal{L}_\pi = \frac{1}{N} \sum_{i=1}^{N} \text{Cross entropy}(\pi(\cdot \mid \phi(x_i^{e_i})), a_i)$$

11:     **for** $i = 1 \dots N$ **do**
12:         $\bar{a}_i \sim \text{Gumbel Softmax}(\pi(\cdot \mid \phi(x_i^{e_i})))$
13:     **end for**
14:     Next state energy loss:

$$\mathcal{L}_{energy} = \frac{1}{N} \sum_{i=1}^{N} E_{\bar{\theta}}(\psi(g_s(\phi(x_i^{e_i}), \bar{a}_i), g_\eta^{e_i}(\mu^{e_i}(x_i^{e_i}), \bar{a}_i)))$$

15:     Parameters update:
16:     $\theta_s \leftarrow \theta_s - \lambda \nabla_{\theta_s}(\mathcal{L}_{inv} + \mathcal{L}_{dyn} + \mathcal{L}_{mi} + \mathcal{L}_\pi)$
17:     $\theta_{g_s} \leftarrow \theta_{g_s} - \lambda \nabla_{\theta_s}\mathcal{L}_{dyn}$
18:     $\theta_\psi \leftarrow \theta_\psi - \lambda \nabla_{\theta_\psi}\mathcal{L}_{dyn}$
19:     **for** $e \in \mathcal{E}_{train}$ **do**
20:         $\theta_\eta^e \leftarrow \theta_\eta^e - \lambda \nabla_{\theta_\eta^e}(\mathcal{L}_{dyn} + \mathcal{L}_{mi})$
21:         $\theta_{g_\eta}^e \leftarrow \theta_{g_\eta}^e - \lambda \nabla_{\theta_{g_\eta}^e}\mathcal{L}_{dyn}$
22:     **end for**
23:     $\theta_\pi \leftarrow \theta_\pi - \lambda \nabla_{\theta_\pi}(\mathcal{L}_\pi + \mathcal{L}_{energy})$

24:     **Environment classifier update:**         ▷ Used to define the invariance loss.
25:     $\mathcal{L}_c = \frac{1}{N} \sum_{i=1}^{N} \text{Cross entropy}(c_s(\phi(x_i^{e_i})), e_i)$
26:     $\theta_c \leftarrow \theta_c - \lambda \nabla_{\theta_c}\mathcal{L}_c$

27:     **Mutual information (MINE) update:**     ▷ Used to define the mutual information loss.
28:     Update $T_{\theta_m}$ by ascending the gradient of the mutual information loss: $\theta_m \leftarrow \theta_m + \nabla_{\theta_m}\mathcal{L}_{mi}$
29: **end while**
30: **Output:** Learnt parameters $\theta_s, \theta_{g_s}, \{\theta_\eta^e, \theta_{g_\eta}^e\}_{e \in \mathcal{E}_{train}}, \theta_\psi, \theta_c, \theta_\pi, \theta_m$

---

# D Causal features for imitation using invariant risk minimization

In the supervised learning setting, Invariant Risk Minimization (IRM) [6] leverage data from multiple domains to learn a data representation $\Phi : \mathcal{X} \to \mathcal{H}$ that elicits an invariant predictor $w : \mathcal{H} \to \mathcal{Y}$ across the different environments, where $\mathcal{X}$ and $\mathcal{Y}$ are the input and output spaces respectively. The training data from each environment $e \in \mathcal{E}$ corresponds to different interventions on the data generating process and $R^e$ corresponds to the empirical risk of the classifier in each domain. We use the following formal definition from [6] to describe the characteristics we want from the learnt representation.

**Definition D.1.** *[6]: We say that a data representation $\Phi : \mathcal{X} \to \mathcal{H}$ elicits an invariant predictor across environments $\mathcal{E}$ if there is a classifier $w : \mathcal{H} \to \mathcal{A}$ simultaneously optimal for all environments, that is, $w = \arg\min_{\bar{w}:\mathcal{H}\to\mathcal{A}} R^e(\bar{w} \circ \Phi)$, for all $e \in \mathcal{E}$.*

Given data from training environments $\mathcal{E}_{train}$, the IRM objective aims to find a representation $\Phi$ such that there exists a classifier $w$ that is optimal across all training domains:

$$\min_{\Phi:\mathcal{X}\to\mathcal{H}, w:\mathcal{H}\to\mathcal{Y}} \sum_{e\in\mathcal{E}_{train}} R^e(\bar{w} \circ \Phi) \text{ subject to } w \in \arg\min_{\bar{w}:\mathcal{H}\to\mathcal{A}} R^e(\bar{w} \circ \Phi), \forall e \in \mathcal{E}_{train} \quad (12)$$

This represents a challenging, bi-level optimization and [6] propose the IRM-v1 objective which is a practical version to optimize:

$$\min_{\phi:\mathcal{X}\to\mathcal{H}} \sum_{e\in\mathcal{E}_{train}} R^e(\Phi) + \lambda_{penalty} \cdot \|\nabla_{w|w=1.0} R^e(w \cdot \Phi)\| \quad (13)$$

Through this optimization, the IRM objective should learn a predictor that only uses the causal parents of the target variable and that is thus invariant across environments. In the supervised setting considered by IRM [6], $R^e$ is the risk of the classifier in environment $e$. For classification and regression problems, this can represent for instance the cross-entropy loss.

**Imitation learning risk:** We propose extending IRM to the imitation learning setting by using instead an imitation risk $R^e$ as described in equation 2. Moreover, in this setting, our aim is to find an invariant policy $\pi$ across the different environments (instead of the invariant classifier $w$). The risk $R^e$ will therefore be specific to the imitation learning algorithm used. For instance, in behaviour cloning (BC) [10], for categorical actions $R^e_{\text{BC}} = \text{Cross entropy}(\pi(\cdot \mid x_t), a_t)$. Alternatively, ValueDice(VDICE) [2] minimizes the disparity between the occupancy measure of the expert policy vs the imitator policy $R^e_{\text{VDICE}} = D_{KL}(\rho^e_D \| \rho_\pi)$. We use the IRMv1 objective in conjunction with the following imitation learning algorithms Behaviour Cloning (BC) [10], Reward-regularized Classification for Apprenticeship Learning (RCAL) [14], ValueDice(VDICE) [2] and Energy-based Distribution Matching (EDM) [3].

| Environments | Original Obs. Space | Action Space | Demonstrator | Random Perf. | Demonstrator Perf. |
|---|---|---|---|---|---|
| Acrobot-v1 | Continuous (6) | Discrete (3) | PPO2 Agent | $-439.92 \pm 13.14$ | $-87.32 \pm 12.02$ |
| CartPole-v1 | Continuous (4) | Discrete (2) | DQN Agent | $19.12 \pm 1.76$ | $500.00 \pm 0.00$ |
| LunarLander-v2 | Continuous (8) | Discrete (4) | PPO2 Agent | $-452.22 \pm 61.24$ | $271.71 \pm 17.88$ |
| BeamRider-v4 | Continuous $(210 \times 160 \times 3)$ | Discrete (9) | PPO2 Agent | $754.84 \pm 214.85$ | $1623.80 \pm 482.27$ |
| MIMIC-III | Continuous (208) | Discrete (2) | Clinician | - | - |

Table 3: Environment details. The random and demonstrator performances are averaged over 1,000 episodes roll-outs.

## E  Experimental details

### E.1  Environments details

We use the following control tasks from OpenAI gym for experiments [47]: Acrobot [49], Cartpole [50], Lunar Lander [47] and BeamRider [51]. For each task, we use pre-trained RL agents from RL Baselines Zoo [52] and Stable OpenAI Baselines [53] to obtain expert policies. We provide in Table 3 details about the different environments used including the size of the observation and action space, the agent used as demonstrator, the demonstrator's performance (average return) as well as the performance of an agent randomly selecting actions. The performance on the OpenAI gym tasks is measured in terms of average return of running the agent in the environment.

Note that the expert uses the original observation space for each task. To create the different environments with spurious correlations used to train and test the imitation learning benchmarks on Acrobot [49], Cartpole [50] and Lunar Lander [47], we augment the observation space as follows. We add 3 noise variables that are different multiplicative factors of the last 3 variables in the original state space. For Acrobot, these are cosine of the second rotational angle and the two joint angular velocities, for CartPole these are Cart Velocity, Pole Angle, Pole Angular Velocity and for LunarLander these are Lander angular velocity, leg 1 ground contact and leg 2 ground contact. The invariant causal state is represented by the original variables in the state space of each control task. We train the benchmarks on demonstrations from two environments with $1\times$ and $2\times$ multiplicative factors for the spurious correlations and we test on an environments with multiplicative factors sampled from $\mathcal{U}(-1, 1)$.

Alternatively, for BeamRider [51], we create the different environments for training and testing by using different camera angles, similarly to [7]. More specifically, we use 2 training environments where the game frames are rotated by 10 degrees to the left and to the right respectively and a test environment that does not have any rotation.

For each OpenAI Gym task, and for each benchmark, we obtain datasets with $N_{traj} \in \{1, 5, 10, 15, 20\}$ demonstrated trajectories from each of the two training environments. We evaluate each benchmark in the test environment by computing the average return over 300 episodes roll-outs. We repeat each experiments 10 times, each time sampling different expert demonstrations for training.

In addition, we also use a dataset from the Medical Information Mart for Intensive Care (MIMIC-III) database [48]. For each patient, we extract 52 clinical covariates including vital signs (e.g. respiratory rate, heart rate, temperature, O2 saturation) and lab test (e.g. glucose, hemoglobin, magnesium, potassium, platelet count, white blood cell count) that are aggregated every hour during their ICU stay. We consider patient trajectories that are up to 24 hours. Moreover, we concatenate the last 4 hours to build the observations received by each imitation learning algorithm. The expert in this case is the doctor and we consider as action the ventilator support. We consider three environment each with 2000 independent patient trajectories from MIMIC III. Two of the environments are used for training and one for testing. We augment the original feature space by adding 20 spurious correlations (noise variables) that are the same as the expert actions with probabilities $p = 0.1$ and $p = 0.2$ in the training environments and with probability $p = 0.8$ in the testing environment.

### E.2 Implementation details

Similarly to [3] and for a fair comparison, whenever possible, we use the same policy network architecture for all imitation learning benchmarks. For Acrobot [49], Cartpole [50], Lunar Lander [47] and MIMIC-III [48] we use a policy network consisting of two fully-connected hidden layers with 64 units each and with ELU activation. Alternatively, for BeamRider [51], we use as the policy network a convolutional neural network with 3 convolutional layers consisting of 32-64-64 filters, followed by a fully connected layer of size 64, with all layers followed by ReLU activations.

We consider discrete actions in all environments; thus, the output layer of the policy network has the same number of dimensions as the action space. For all the different environments used for evaluation we optimize the parameters using the Adam Optimizer for 10k iterations with learning rate $\lambda = 0.001$ and batch size 64 [3]. Moreover, we use the publicly available code for the different benchmarks used and other than the standardized policy network, we keep the optimal hyperparameters in the original implementations.

The experiments were run on a system with 6CPUs, an Nvidia K80 Tesla GPU and 56GB of RAM.

**Invariant Causal Imitation Learning:** uses a policy network as described above and neural network architectures with two fully connected hidden layers with 64 units and with ELU activation for each of $\phi$, $\mu^e$, $g_s$, $g_\eta$, $\psi$, $c_s$ and $T_{\theta_m}$ in the Acrobot [49], Cartpole [50], Lunar Lander [47] and MIMIC-III [48] experiments. On the other hand, for the BeamRider environment [51], we use a convolutional neural network with 3 convolutional layers consisting of 32-64-64 filters, followed by a fully connected layer of size 64, with all layers followed by ReLU activations for $\phi$ and $\mu^e$. Moreover, we use neural network architectures with two fully connected hidden layers with 64 units and with ELU activation for $g_s$, $g_\eta$, $c_s$, $T_{\theta_m}$ and for the policy network $\pi$. Finally, for $\psi$ we use a neural network with 2 fully connected layers of 64 and $64 \times 7 \times 7$ hidden units followed by 3 transposed convolution layers consisting of 64-64-32 filters.

**Behaviour cloning (BC):** We implement behaviour cloning by using a policy network architecture as described above. We train the model using cross entropy loss and we optimize it as described above. We use the same hyperparameters for BC-IRM but instead use IRM-v1 objective.

**Reward-regularized Classification for Apprenticeship Learning (RCAL):** We implement RCAL by adding a sparsity-based loss on the implied rewards [14] and we set the sparsity-based regularization coefficient to 0.01. We use the same policy network architecture for optimization procedure as described above. Moreover, we use the same hyperparameters for RCAL-IRM but instead use IRM-v1 objective.

**Energy-based Distribution Matching (EDM):** We use the publicly available implementation for EDM [59] from here: `https://github.com/vanderschaarlab/mlforhealthlabpub`. We use the same policy network and optimization procedure as above, which corresponds to the ones used in [59]. Moreover, following the implementation details provided in [59] we set the joint EBM training hyperparameters to noise coefficient $\sigma = 0.01$, buffer size $\kappa = 10000$, length $l = 20$, re-initialization $\delta = 0.05$ and SGLD step size $\alpha = 0.01$. For EDM-IRM we use the same hyperparameters, but instead optimize the IRM-v1 objective.

**ValueDice (VDICE):** We use the publicly available implementation for VDICE [2] from `https://github.com/google-research/google-research/tree/master/value_dice`. However, to adapt the model to discrete actions we modify the last layer of the actor network to use Gumbel-softmax. For Acrobot [49], Cartpole [50], Lunar Lander [47] and MIMIC-III [48], the actor and discriminator network architecture used have two fully connected hidden layers with 64 units and ReLU activation. Conversely, for BeamRider [51] the actor and discriminator networks consist of 3 convolutional layers consisting of 32-64-64 filters, followed by a fully connected layer of size 64, with all layers followed by ReLU activations. As described in [2], orthogonal regularization is used for the actor and a learning rate of 0.00001. The discriminator uses a learning rate of 0.001. For VDICE-IRM we use the same hyperparameters, but instead optimize the IRM-v1 objective.

# F Additional experiments

## F.1 Ablations

To understand the impact of the different components in the overall loss function used to train ICIL, we performed an ablation experiment on the CartPole [50] control task from OpenAI gym [47]. We follow the same training and testing set-up described in Section 5.1.

Let $L = L_\pi + L_{dyn} + L_{inv} + L_{mi} + L_{energy}$ be the full loss function used for training ICIL. Refer to Section 4.1 for details of how each component in $L$ is defined. The results in Figure 5 illustrate the impact of removing different terms from this loss function on overall performance (average return of the learnt imitation policy in the test environment). The average return is scaled between 1 (expert performance) and 0 (random policy performance). The setting of only using $L_\pi$ corresponds to the Behaviour Cloning (BC) benchmark.

We notice that while each term in the loss $L$ used to train ICIL is important for the overall performance, the loss term $L_{inv}$ which ensures that the state representation is invariant across environments plays the most significant role on the performance in the test environment. This is due to the fact that $L_{inv}$ is crucial for learning the shared latent structure across the different environments that consists of the causal parents of the actions.

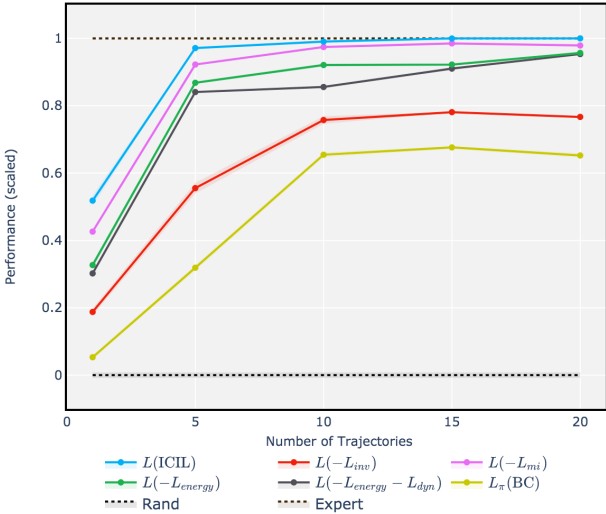

Figure 5: Evaluating the impact of the different loss components on overall performance for the CartPole control task. $x$-axis indicates the number of trajectories (in $\{1, 5, 10, 15, 20\}$) with expert demonstrations from each training environment given as input to each ablated version of ICIL and $y$-axis represents average return of the learnt imitation policy on the test environments, scaled between 1 (expert performance) and 0 (random policy performance).

## F.2 Train vs. test performance

We report here the evaluation metrics for both training and testing environments on the CartPole [50] control task. We follow the same experimental set-up described in Section 5.1. In Figure 6 we report the performance of ICIL and BC when evaluated both on 300 new episode roll-outs from one of the environments used for training as well as when evaluated on a test environment with different multiplicative factors for the noise variables. We notice that the imitation policy learnt by BC relies on the spurious correlations and thus fails to generalize beyond the environments it has been exposed to. On the other hand, ICIL learns an imitation policy that correctly depends on the state variables, which are the ones that are being shared across the different environments.

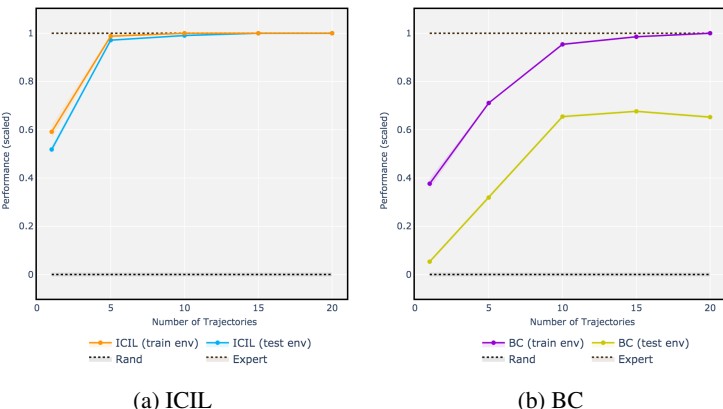

(a) ICIL            (b) BC

Figure 6: Train vs. test performance on CartPole. $x$-axis indicates the number of trajectories (in $\{1, 5, 10, 15, 20\}$) with expert demonstrations from each training environment given as input to each benchmark and $y$-axis represents average return of the learnt imitation policy when evaluated on both the train and test environments, scaled between 1 (expert performance) and 0 (random policy performance).

### F.3 Robustness to increasing the size of spurious correlations

Finally, we investigate the robustness of the different imitation learning methods to increasing the size of the spurious correlations (i.e. the number of noise variables in each environment). In this case, we consider again the CartPole [50] control task and the setting where 5 trajectories from each training environment are given as input to each benchmark during training. The spurious correlations in each environment are different multiplicative factors of the last 3 variables in the original state space of the control task. We follow the same set-up described in Appendix for setting the multiplicative factors for the train and test environments. Figure 7 illustrates the performance of ICIL, BC and EDM when increasing the number of noise variables used for the different environments. We notice that ICIL is robust to having more spurious correlations, while the performance of BC and EDM degrades more significantly in the case where the observations from the expert demonstrations have a large number of noise variables.

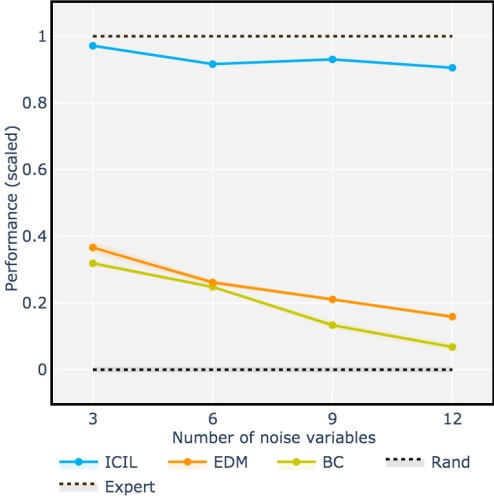

Figure 7: Robustness to increasing the number of noise variables. $x$-axis indicates the number of noise variables (in $\{3, 6, 9, 12\}$) that are part of the observations in each environment and $y$-axis represents average return of the learnt imitation policy on the test environments, scaled between 1 (expert performance) and 0 (random policy performance).