# OpenReview forum: "Invariant Causal Imitation Learning for Generalizable Policies"
_NeurIPS.cc/2021/Conference — NeurIPS 2021 Poster_

### Official Review · Reviewer_YxyH · 2021-07-16

**Rating:** 6
**Confidence:** 4

**Summary:**

The paper presents an imitation learning approach that can generalize across demonstrations acquired from multiple environments. The approach projects the observations in an invariant shared latent space across environments. The variability across environments is modeled as noisy perturbations to reflect spurious correlations with the actions. The noisy perturbations and invariant state representations are learned such that the transition dynamics are preserved, while the mutual information between them is minimized. Experiments on OpenAI gym benchmarks and MIMIC III derived dataset suggest performance improvement over other competing methods.

**Limitations And Societal Impact:**

see above. No negative societal impact.

**Main Review:**

The paper is well-written for the most part. The problem formulation is interesting and the rationale for the choice of loss functions is well-motivated. Fig. 3 is helpful to explain the ideas of the paper. Experimental results are sound, and interesting for a wider imitation learning community.

- The problem of generalization in imitation is studied in several different contexts. Environments can vary across pixel distributions (such as sim or real [1]), dynamics [2], reward functions and so on. It would be useful to investigate these cases independently, and in combination with each other. For example, situations where the dynamics and the reward function are the same but pixel distributions are different, does \eta^e converge to a trivial solution ?

- As the paper addresses dynamics invariance, it would be useful to specify the generalization context clearly (including the title).

- Transfer learning and domain generalization are closely related to the proposed approach. Related work can be better grounded in literature, in particular with [7]. I am also not convinced if the paper is the first to tackle generalizable imitation learning in a batch setting.

- It would be useful to report ablation studies. While the loss functions are well-motivated, they likely have competing objectives which can make the learning unstable. Results reporting the role of each loss function alone and in combination would give furhter insights about the efficacy of different terms.

- Generalizing across different environments here means doing well on average, but the resulting imitation behvaior may not be optimal for a particular environment. Additional experiments to evaluate the imitation performance in a single environment vs multiple environments as a function of learned \eta would be helpful.

[1] DIRL: Domain-Invariant Representation Learning for Sim-to-Real Transfer, 2020

[2] Off-Dynamics Reinforcement Learning: Training for Transfer with Domain Classifiers, 2020-21


**Time Spent Reviewing:**

3

---

> ### Author Response · Authors · 2021-08-10
> **Response to Reviewer YxyH [Part 2/2]**
>
> **Ablation studies**
>
> We agree with the reviewer that ablation studies are needed to understand the benefit of each loss function. we have performed additional experiments on the Atari BeamRider environment where, similarly to [7], different camera angles are used for the training and testing environments. In particular, we use two training environment where the game frames are rotated by 10 degrees to the left and to the right respectively, while the test environment has no rotation.  The rotation is applied to the entire frame for all trajectories in each environment. However, note that, despite the rotation, the dynamics for the state variables $s_t$ and how they influence the action through $p(a_t | s_t)$ stay the same.
>
> In this setting, we have performed ablation studies to evaluate the impact of the different loss functions. Let $L = L_{pi} + L_{dyn} + L_{inv} + L_{mi} + L_{energy}$ be the full loss function. We evaluated the impact on performance of removing different terms from this loss function:
>
> - $L$: 0.78 ± 0.06
> - $L (- L_{inv})$: 0.65 ± 0.05
> - $L (- L_{mi})$: 0.75 ± 0.04
> - $L (- L_{energy})$: 0.73 ± 0.05
> - $L (- L_{energy} - L_{dyn})$: 0.70 ± 0.06
> - $L (- L_{dyn} - L_{inv} - L_{mi} -L_{energy})$: 0.59 ± 0.04
>
> We will include results for the ablation studies across different number of trajectories in the revised manuscript.
>
>
> **Generalization and doing well on average**
>
> In our current experiments, the expert performance is the optimal performance for the test environment. We note that in the Acrobot and CartPole environment, the learnt imitation policy from ICIL reaches expert performance on the test environment, while on LunarLandar it reaches 82% of expert performance.
>
> To further address your comment, we will also report in the revised manuscript both the evaluation on train and test environments to show the generalization gap for standard batch imitation learning benchmarks. In particular, the *degradation* between the two measures will quantify the (in-)ability of a method in generalizing beyond the environments it has been exposed to. We will report both train and test performance in the revised manuscript for all experiments. In the following, we give an example on BeamRider for five trajectories.
> Consider evaluating a method after training on the batch data. For BC, evaluating its performance on the training environments yields a score of 0.66 ± 0.05, whereas evaluating its performance on the test environment (i.e. unseen environment) yields a score of 0.59 ± 0.04. *The degradation is ~0.07.* Contrast this to our model, where evaluating its performance on the training environments yields a score of 0.82 ± 0.06, and evaluating its performance on the test environment yields a score of 0.78 ± 0.06. *The degradation is approximately half that of BC.*
>
> Moreover, we are now conducting additional experiments to evaluate the impact of increasing the number of noise variables and of increasing the magnitude of the noise on the model’s performance. We will also add evaluation on a larger number of test environments.
> Please note that as we add more noise variables, we expect the gap between the standard imitation learning methods (BC, EDM, RCAL, VDice) and our ICIL method to increase.

---

> > ### Comment · Reviewer_YxyH · 2021-08-31
> > **response to authors**
> >
> > Thank you for the detailed response !
> >
> > Appreciate your input on clarifying the concerns as well as conducting ablation studies to ground the claims. Looking forward to seeing more of the experiments in the final version. It would be useful to comment that the studied setting is closest to "domain generalization" where there are multiple environments and the observations may change across multiple environments, but latent dynamics remain the same.

---

> ### Author Response · Authors · 2021-08-10
> **Response to Reviewer YxyH [Part 1/2]**
>
> Thank you very much for your thoughtful comments and feedback!
>
> **Problem of generalization in imitation**
>
> Thank you for your suggestion! Please note that we are already considering the case where the dynamics of the noise variables are varying, but the dynamics of the state variables are invariant. This for instance would represent the scenario suggested by the reviewer where the pixel distributions are different between the environments, but the dynamics are the same. In this case, $\eta$ will encode these differences in pixel distribution. While our paper does not tackle the case where the full environment dynamics are changing between training and testing, we believe this also represents an important scenario to consider and we will mention it as future work. We will also include the references provided by the reviewer to the related works and compare them with our approach.
>
> **Dynamics invariance**
>
> Thank you for your suggestion! We will further clarify the generalization context. Please note that Figure 1 illustrates the type of dynamics invariances we assume. In particular, we only assume that $p(s_{t+1} \mid s_t, a_t)$ is invariant across environments and we allow $p(\eta_{t+1} \mid \eta_t, a_t)$ to stay the same.
>
>
> **Clarifications and differences from transfer learning and domain generalization**
>
> We agree with the reviewer that the problem context may benefit from clarification.
>
> Let us first clarify the distinctions between our setting, domain adaptation, and transfer learning as they are (commonly) used. Suppose we have inputs $x$ and outputs $y$:
> - Domain adaptation: $p(x)$ changes. We are given samples from the original domain $p(x)$, and some from a new domain $p^{\prime}(x)$ that we wish to adapt to, in order to still predict $y$ given $x$ well.
> - Transfer learning: In addition to the distribution $p(x)$ that may possibly change, here $p(y|x)$ may also change. As before, we wish to transfer learning from data from $p(y|x)$ to data from some other $p^{\prime}(y|x)$.
> - Generalization: In the context of either of the above kinds of shifts, “generalization” usually means we are *not* given in advance specific knowledge/data about the new distributions that we will encounter.
> - Invariant learning: This is the setting we operate in. We are given data from more than one domain, but here the purpose is to utilize this to better *generalize* to unseen domains.
>
> To be clear, in our setting with multiple environments,
> - $p(x)$ is free to differ between them, and
> - $p(a|x)$ is free to differ between them.
>
> The main assumption we make (3.1) is that the environments and tasks are the same modulo noise, i.e. that there exists some non-empty $\mathcal{S}$ such that:
> - $p(s)$ is the same between them, and
> - $p(a|s)$ is the same between them.
>
> And to ensure that this space can actually be learned, a secondary assumption (3.2) is that there exists some non-empty $\mathcal{Z}$ that differs across the environments.
>
> We allow different environments to have different structure by having different $p(x)$ marginals (as well as different $p(a|x)$). The only requirement is that the environments are the same *as far as the task is concerned*. This means that there exists some $\mathcal{S}$ such that $p(s)$ marginals should be the same (as well as $p(a|s)$). Learning such a representation $s = \phi(x)$ that is invariant satisfies the standard Environment Invariance Constraint [R1].
>
> In other words, we have a set of environments that are different (i.e. the dynamics of $x_t$ are different), but the task being performed by the agent is the same (i.e. the dynamics of $s_t$ are the same). This setting applies to e.g. lightning conditions in a room changing, but physical dynamics of the environment staying the same [7]; weather conditions changing, but driving behaviour and dynamics staying the same.
>
> [R1] Creager, Elliot, Jörn-Henrik Jacobsen, and Richard Zemel. "Environment inference for invariant learning." International Conference on Machine Learning. PMLR, 2021.

---

### Official Review · Reviewer_raJe · 2021-07-16

**Rating:** 7
**Confidence:** 3

**Summary:**

This paper proposes a method for performing imitation learning from a set of expert demonstrations collected across a set of environment variations, where each variation is an intervention on the noise variables of the environment. The method, ICIL, is demonstrated to successfully control for spurious correlations in a set of OpenAI Gym envronments and a healthcare domain dataset.

**Limitations And Societal Impact:**

- The method makes the assumption that the environments from which demonstrations are collected correspond to interventions only on the noise variables of the environment. It would be valuable for the authors to describe real world scenarios meeting these assumptions, or if such examples are hard to come by, describe how these assumptions are relaxed in example practical domains.

**Main Review:**

The method itself is well motivated with a clear formalization of the problem setting. The work is well positioned with respect to related works, and the specific problem of learning a generalizable policy from offline data is an important one.

My main concern relates to how the experiments focus on highly contrived toy cases specifically designed to fit the formal assumptions of their method. It would be beneficial to include experiments that test the method in settings where the examples were not hand-designed by the authors, e.g. OpenAI Procgen, where many levels show differences that can be viewed as spurious noise, e.g. background colors; or a version of Mujoco where the background changes across episodes. These more challenging examples would test how well this method can scale in practice, and how robust this method is to domains that do not perfectly match the structural assumptions around per-environment interventions only occuring on noise variables.

If the authors provide results on a less contrived domain, I will gladly raise my score to at least 7.

### Post-rebuttal update:
The authors' rebuttal addresses my concerns, so I am raising my score to a 7.

**Time Spent Reviewing:**

5

---

> ### Author Response · Authors · 2021-08-10
> **Response to Reviewer raJe**
>
> Thank you very much for your thoughtful comments and feedback!
>
> **Additional experiments on Atari BeamRider**
>
> Thank you for your suggestion! We have now performed additional experiments on BeamRider where, similarly to [7], different camera angles are used for the training and testing environments. In particular, we use two training environments where the game frames are rotated by 10 degrees to the left and to the right respectively, while the test environment has no rotation. Note that these rotations are applied to the full frame of the Atari game.  However, note that, despite the rotation, the dynamics for the state variables $s_t$ and how they influence the action through $p(a_t | s_t)$ stay the same.
>
> For the setting with 5 trajectories with expert demonstrations from each training environment given as input to each benchmark, we report the average return of the learned imitation policy on the test environment, scaled by the expert performance.
> - BC: 0.59 ± 0.04
> - BC-IRM: 0.62 ± 0.06
> - EDM: 0.64 ± 0.05
> - EDM-IRM: 0.66 ± 0.05
> - ICIL: 0.78 ± 0.06
>
> We will include in the revised manuscript results across {1, 5, 10, 15, 20} trajectories for the training environment and across all benchmarks. Due to time limitations, we have select to report here results for behaviour cloning (BC), the standard imitation learning benchmark and EDM, which achieves best performance overall in our other experiments among the benchmarks considered.
>
> **Limitations And Societal Impact**
>
> Thank you for your suggestion to include additional discussion about real-world scenarios satisfying our assumptions! Several examples where our setting is applied in practice are for instance lightning conditions (noise variable) in a room changing, but physical dynamics of the environment staying the same [7]; weather conditions changing, but driving behaviour and dynamics staying the same. Such environments only have interventions on the noise variables.
> To further clarify our assumptions, in our setting with multiple environments,
> - $p(x)$ is free to differ between them, and
> - $p(a|x)$ is free to differ between them.
>
> The main assumption we make (3.1) is that the environments and tasks are the same modulo noise, i.e. that there exists some non-empty $\mathcal{S}$ such that
> - $p(s)$ is the same between them, and
> - $p(a|s)$ is the same between them.
>
> And to ensure that this space can actually be learned, a secondary assumption (3.2) is that there exists some non-empty $\mathcal{Z}$ that differs across the environments.
>
> In other words, we have a set of environments that are different (i.e. the dynamics of $x_t$ are different), but the task being performed by the agent is the same (i.e. the dynamics of $s_t$ are the same).
>
> We will further discuss these assumptions and give additional examples of real world scenarios meeting these assumptions in the revised paper.

---

> > ### Comment · Reviewer_raJe · 2021-08-17
> > **Thanks for the additional experiments and explanation**
> >
> > I appreciate your running additional experiments per my feedback, and further expounding on real-world settings in which your method may be applicable. These responses address my critique, so I am raising my score to a 7.

---

### Official Review · Reviewer_VV1t · 2021-07-17

**Rating:** 7
**Confidence:** 4

**Summary:**

The authors consider the setting of batch imitation learning with access to expert demonstrations over multiple environments which correspond to interventions on shared, underlying causal structure. They use the framework from [7] to propose a method that disentangles the relevant and irrelevant components of the state for a task with an additional regularization component that minimizes the mutual information between the two components with MINE. For the imitation policy, they propose an energy-based model approach to approximate the occupancy measure of the expert policy as a way to regularize and train the policy.


**Limitations And Societal Impact:**

Yes

**Main Review:**

At first glance imitation learning in the batch setting seems like a strange setup, and therefore not so unusual that it is original. However, the authors motivate this setting well by pointing out that in the medical domain, these are reasonable assumptions to make — that you are in an offline setting with access to expert data from different environments with shared latent structure.

As the authors themselves point out, the setup, assumption, and parts of the method are directly from [7] but there are novel components to modify it for the batch and imitation learning setting. The experiments seem reasonable, although the unusual metric on the Open AI Gym tasks make it difficult to ground these results to those reported in other works.  Perhaps a more relevant benchmark would be D4RL, which has an expert dataset.

The MIMIC III experiment, given the setting, is well chosen. The results there also are clearly quite significant and with respect to reasonable baselines. It would also perhaps be best to report both train and evaluation performance given the emphasis on the focus being generalization.

Overall this seems like a solid contribution and I would lean towards accept.

**Time Spent Reviewing:**

3

---

> ### Author Response · Authors · 2021-08-10
> **Response to Reviewer VV1t**
>
> Thank you very much for your thoughtful comments and suggestions! We will include a reference to the D4RL benchmark.
>
> Moreover, we have performed additional experiments on the Atari BeamRider environment where, similarly to [7], different camera angles are used for the training and testing environments. In particular, we use two training environments where the game frames are rotated by 10 degrees to the left and to the right respectively, while the test environment has no rotation.  The rotation is applied to the entire frame for all trajectories in each environment. However, note that, despite the rotation, the dynamics for the state variables $s_t$ and how they influence the action through $p(a_t | s_t)$ stay the same.
>
> For the setting with 5 trajectories with expert demonstrations from each training environment given as input to each benchmark, we report the average return of the learnt imitation policy on the test environment, scaled by the expert performance.
>
> - BC: 0.59 ± 0.04
> - BC-IRM: 0.62 ± 0.06
> - EDM: 0.64 ± 0.05
> - EDM-IRM: 0.66 ± 0.05
> - ICIL: 0.78 ± 0.06
>
> We will include in the revised manuscript results across {1, 5, 10, 15, 20} trajectories for the training environment and across all benchmarks. Due to time limitations, we have selected to report here results for behaviour cloning, the standard imitation learning benchmark and EDM, which achieves best performance overall in our other experiments among the benchmarks considered.
>
> **Evaluation Metrics for Both *Training* and *Testing* Environments**
>
> Thank you for the suggestion to include evaluation metrics for both *training* and *testing* environments. In particular, the *degradation* between the two measures will quantify the (in-)ability of a method in generalizing beyond the environments it has been exposed to. We will report both train and test performance in the revised manuscript for all experiments. In the following, we give an example on BeamRider for five trajectories.
>
> Consider evaluating a method after training on the batch data. For BC, evaluating its performance on the training environments yields a score of 0.66 ± 0.05, whereas evaluating its performance on the test environment (i.e. unseen environment) yields a score of 0.59 ± 0.04. *The degradation is ~0.07.* Contrast this to our model, where evaluating its performance on the training environments yields a score of 0.82 ± 0.06, and evaluating its performance on the test environment yields a score of 0.78 ± 0.06. *The degradation is approximately half that of BC.*
>
> We will report this gap between training and testing performance in the revised manuscript for all models.

---

> > ### Comment · Reviewer_VV1t · 2021-08-17
> > **Response to rebuttal**
> >
> > Thanks for giving these numbers and running evaluation on test environments for BeamRider. Looking forward to seeing the results across more experiments! Given these promising results, I still advocate to accept this paper.

---

### Official Review · Reviewer_GSwe · 2021-07-17

**Rating:** 7
**Confidence:** 3

**Summary:**

This paper proposes a new imitation learning algorithm, ICIL, which aims to make imitation learning more robust to irrelevant distractor features (i.e. features that are not causally upstream of the chosen expert action across a range of environments). Starting from BC, ICIL first adds a state encoder that splits states into an action-relevant component $s_t$ and a noise/distractor component $\eta_t$. Next it adds three additional auxiliary losses to prevent distractors from influencing $s_t$: a "dynamics-preserving" loss to ensure that $(s_t, \eta_t)$ are sufficient to predict the next environment observation; an "environment invariance" loss to make $s_t$ marginally independent of the environment; and a mutual information loss that regularises $s_t$ and $\eta_t$ towards marginal independence. Finally, ICIL incorporates an auxiliary loss to match next-step expert state occupancy under a learnt dynamics model, with the intent of decreasing compounding error. Experiments show that ICIL gets higher return than several baselines in variants of standard benchmarks that have been adapted to include additional distractor features.

**Limitations And Societal Impact:**

Limitations and societal impact were discussed adequately.

**Main Review:**

In my view, the main strengths of this paper are:

1. Clarity. Writing quality was high on the whole: the problem is motivated in a compelling way, assumptions are clearly stated, and description of the algorithm and experiments makes sense. My only objection is to the assumptions and desiderata that are introduced without enough justification in sections 3-4 (see weaknesses below).
2. The experiments have a wide selection of baselines. Incorporating IRM into the baselines was particularly appreciated—it's hard to find appropriate baselines for a problem like this, so it's good to see some novel baselines that at least try to solve the same problem in different ways. Moreover, the proposed algorithm manages to improve substantially over the baselines in all the examined environments.

However, there are also some weaknesses to the algorithm:

1. The algorithm makes several non-trivial assumptions and has many moving parts, and some of them seemed arbitrary or poorly motivated. For instance, it's assumed that the transitions of decision-relevant variables are the same between "environments", but transitions of noise variables are not. What kinds of domains is this true for in practice? The desiderata in Section 4.1 (invariance across environments, dynamics-preserving $s$/$\eta$, disentangled $s$/$\eta$) are also vague and get introduced without justification:
   1. Why is having equal $s_t$ marginals across environments an adequate notion of "invariance"? What about environments where the demonstrator really does have different $s_t$ marginals because of some difference in environment structure? It seems like $\mathcal L_{\text{inv}}$ objective is going to throw away some relevant information in such cases.
   2. Having "dynamics-preserving" state representations seems too weak—why not just have an invertible mapping from $x$ to $(s,\eta)$? Such a mapping would still be "dynamics-preserving" while also satisfying Assumption 3.1 directly.
   3. What does it mean for $s$ and $\eta$ to be "disentangled"? Why is this desirable? "Disentangled" is a loaded word, and statistical independence (which an MI penalty incentivises) is neither necessary nor sufficient to obtain the kind of disentanglement that you see in carefully hand-designed state representations (for a counterexample, see Theorem 1 of "Challenging Common Assumptions in the Unsupervised Learning of Disentangled Representations" by Locatello et al.).
2. I also had several concerns about experiments:
   1. The various losses (there are five of them!) do not seem to have been examined separately in an ablation, so it's not clear which are responsible for the improvement over baselines.
   2. The experiment domains all have manually-inserted distractors that are deliberately constructed to meet the assumptions of the algorithm. The experiments thus do not provide evidence that the assumptions are adequate to handle the sort of distractors present in real-world tasks.
   3. The experiments report return/accuracy, but do not directly evaluate how sensitive the various models are to the inserted distractor features. In some ways this would be more informative than just looking at overall performance, since it would distinguish between gains from reduced reliance on distractors and gains from improved optimisation, etc.

On the whole I feel that this is a well-written paper, but I am currently ambivalent about acceptance because I do not feel the justification and evaluation of the algorithm is adequate given how complex the algorithm is. Clarifying points raised in the weaknesses above would help.

**Time Spent Reviewing:**

3.5

---

> ### Author Response · Authors · 2021-08-10
> **Response to Reviewer GSwe [Part 3/3]**
>
> **(2) Details on Experiments**
>
> **2.1/2 Ablations and More Complex Environment**
>
> Thank you for your comments about the experiments. We agree with the reviewer that it is important to perform ablation studies and to evaluate the model in a more complex environment.
>
> Therefore, we have performed additional experiments on the BeamRider environment where, similarly to [7], different camera angles are used for the training and testing environments. In particular, we use two training environments where the game frames are rotated by 10 degrees to the left and to the right respectively, while the test environment has no rotation.  The rotation is applied to the entire frame for all trajectories in each environment. However, note that, despite the rotation, the dynamics for the state variables $s_t$ and how they influence the action through $p(a_t | s_t)$ stay the same.
>
> For the setting with 5 trajectories with expert demonstrations from each training environment given as input to each benchmark, we report the average return of the learnt imitation policy on the test environment, scaled by the expert performance.
>
> - BC: 0.59 ± 0.04
> - BC-IRM: 0.62 ± 0.06
> - EDM: 0.64 ± 0.05
> - EDM-IRM: 0.66 ± 0.05
> - ICIL: 0.78 ± 0.06
>
> We will include in the revised manuscript results across {1, 5, 10, 15, 20} trajectories for the training environment and across all benchmarks. Due to time limitations, we have select to report here results for behaviour cloning (BC), the standard imitation learning benchmark and EDM, which achieves best performance overall in our other experiments among the benchmarks considered.
>
> In addition, we have also performed ablation studies to evaluate the impact of the different loss functions. Let $L = L_{pi} + L_{dyn} + L_{inv} + L_{mi} + L_{energy}$ be the full loss function. For the same setting considered in the experiment above, we evaluated the impact on performance of removing different terms from this loss function:
>
> - $L$: 0.78 ± 0.06
> - $L (- L_{inv})$: 0.65 ± 0.05
> - $L (- L_{mi})$: 0.75 ± 0.04
> - $L (- L_{energy})$: 0.73 ± 0.05
> - $L (- L_{energy} - L_{dyn})$: 0.70 ± 0.06
> - $L (- L_{dyn} - L_{inv} - L_{mi} -L_{energy})$: 0.59 ± 0.04
>
> We will include results for the ablation studies across different number of trajectories in the revised manuscript.
>
> **2.3 Sensitivity to Distractors**
>
> Thank you for your suggestion! We agree with the reviewer that it is important to consider the sensitivity of the model to the distractor factor.  We are now running additional experiments to evaluate the impact of increasing the number of noise variables and of increasing the magnitude of the noise on the model’s performance. Please note that as we add more noise variables, we expect the gap between the standard imitation learning methods (BC, EDM, RCAL, VDice) and our ICIL method to increase.

---

> > ### Comment · Reviewer_GSwe · 2021-08-26
> > **Combined response**
> >
> > Thank you for the detailed response! I'm increasing my score to a 7, mostly due to the new ablations/experiments, and because I'm more convinced that the environment setup (with separated noise/state) is useful. Here are some more detailed responses:
> >
> > - (1)/1.1: Okay, I'm more convinced of the value of this setup now. It seems somewhat analogous to [block MDPs](https://arxiv.org/pdf/1901.09018.pdf) (defined in section 2.1 of linked paper) in that it's mostly about adapting to different state ($s$) to observation ($x$) mappings. When I said that it does not account for generalisation across different structures, I meant that it does not account for generalisation across different transition dynamics or initial distributions of the latent $s$, which it sounds like the authors agree with. I somewhat disagree with the procgen example—moving backgrounds definitely fall into the category of irrelevant noise, but different seeds in procgen can also yield wildly different "latent MDPs" (in the $\mathcal S$ space), even after taking out decision-irrelevant stuff like the background.
> > - 1.2: This sounds reasonable.
> > - 1.3: I don't think I was confused here—I definitely interpreted the original paper as saying that $s$ and $\eta$ were "disentangled" from _each other_, not that they were vectors with internally disentangled components. My objection is that enforcing marginal independence does not necessarily yield any kind of semantically meaningful "disentanglement"—the theorem in the paper I referenced (Locatello et al.) still applies to pairs of random vectors, and not just components of an individual random vector. This is mostly a disagreement about framing; I'm fine if the paper only claims approximate marginal independence (which is what equation (7) is actually regularising towards) and not disentanglement (which is not even a well-defined term). I've written a bit more about this below.
> >   - As an aside, I think $\psi$ should be $\mu$ in equation (7)—$\psi$ seems to map $\eta$s to $x$s, but here the context implies that it is getting applied to an $x$ and producing an $\eta$.
> > - 2.1/2: Thank you for running these additional experiments! This addresses my objection regarding ablations.
> > - 2.3: Thanks again for running these experiments. I think that direct evidence of lower sensitivity to distractor features will do a much better job of demonstrating that the observed gains really do come from acquiring useful invariances.

---

> ### Author Response · Authors · 2021-08-10
> **Response to Reviewer GSwe [Part 2/3]**
>
> **1.1 Notion of Invariance**
>
> To be clear, we allow different environments to have different $p(x)$ marginals (as well as different $p(a|x)$). This allows environments to have different structure. The only requirement is that the environments are the same *as far as the task is concerned*. This means that there exists some $\mathcal{S}$ such that $p(s)$ marginals should be the same (as well as $p(a|s)$). Learning such a representations $s = \phi(x)$ that is invariant satisfies the standard Environment Invariance Constraint [R2].
>
> Moreover, note that the setting described by the reviewer where the dynamics structure of $s_t$ changes across environments is not considered in our paper. We assume that the dynamics for $s_t$ are invariant across environments. One such example where the environment dynamics are changing is for instance when an agent plays a game, e.g. Atari, on different levels of difficulty and the transition dynamics of the latent state are changing between the different levels of difficulty. In this case, there would be no invariant $\mathcal{S}$ such that $p(s)$ and $p(a|s)$ are invariant across the environments. We have now further clarified this distinction in the problem formalism in Section 3.2.
>
> **1.2 Invertible Mapping**
>
> Thank you for your suggestion! We agree with the reviewer that directly building an invertible mapping from $x$ to $(s, \eta)$ would also be possible. However, please note that the motivation for decoding $s_{t+1}$ and $\eta_{t+1}$ into $x_{t+1}$ is twofold. In addition to learning dynamics-preserving representations, this also allows us to compute the energy of the next state obtained by following the imitation policy and enforcing this to be similar to the distribution of states visited by the expert’s policy. We have now further clarified these considerations in Section 4.1.
>
> **1.3 Meaning of “Disentangled”**
>
> Thank you for your clarifying question. Actually, when we mentioned that we want for s and $\eta$ to be disentangled, we meant that $s$ and $\eta$ should individually be independent from each other---and *not* that the different dimensions in $s$ and $\eta$ to be disentangled from each other. The reason why it is important for the $s$ and $\eta$ representations to be independent is because we do not want to condition the policy on any noise factors.
>
> We agree that the wording may have given the impression that we meant the latter, when we meant the former. We have now revised the wording when introducing this notion in Section 4.1 to eliminate any confusion.
>
>
> [R2] Creager, Elliot, Jörn-Henrik Jacobsen, and Richard Zemel. "Environment inference for invariant learning." International Conference on Machine Learning. PMLR, 2021.

---

> ### Author Response · Authors · 2021-08-10
> **Response to Reviewer GSwe [Part 1/3]**
>
> Thank you very much for your thoughtful comments and feedback! Please find below the answers to your comments and please let us know if you have any additional questions or if there are any points we can further clarify.
>
> **(1) Problem Context and Assumptions**
>
> We agree that the problem context may benefit from clarification. Actually, the setting from our paper is standard in the literature [7], and valid for a variety of common domains in the MDP setting. To be clear, in our setting with multiple environments,
>
> - $p(x)$ is free to differ between them, and
> - $p(a|x)$ is free to differ between them.
>
> The main assumption we make (3.1) is that the environments and tasks are the same modulo noise, i.e. that there exists some non-empty $\mathcal{S}$ such that
>
> - $p(s)$ is the same between them, and
> - $p(a|s)$ is the same between them.
>
> And to ensure that this space can actually be learned, a secondary assumption (3.2) is that there exists some non-empty $\mathcal{Z}$ that differs across the environments.
> In other words, we have a set of environments that are different (i.e. the dynamics of $x_t$ are different), but the task being performed by the agent is the same (i.e. the dynamics of $s_t$ are the same). This setting applies to e.g. lightning conditions in a room changing, but physical dynamics of the environment staying the same [7]; weather conditions changing, but driving behaviour and dynamics staying the same.
>
> Another concrete environment that would satisfy these conditions is the OpenAI Procgen benchmark which has different backgrounds across the levels. It has also been shown in reinforcement learning that when the training environments contain spurious correlations that are not present at test time, the agent’s performance will drop significantly [R1]. Our work aims to address these problems in the imitation learning set-up to learn policies that only depend on the invariant latent structure and that generalize to changes in the noise variables across environments.
>
> To show the applicability of your method in these types of environments, we also evaluate on OpenAI Atari BeamRider environment where we change the camera rotation angle between training and testing environments (see below).
>
> [R1] Zhang, C., Vinyals, O., Munos, R., & Bengio, S. (2018). A study on overfitting in deep reinforcement learning. arXiv preprint arXiv:1804.06893.

---

> ### Author Response · Authors · 2021-08-20
> **Follow-up**
>
> Dear reviewer,
>
> Thank you once again for your useful comments and invaluable feedback on our paper! Please let us know if our response has addressed your concerns. If you have any additional comments, please let us know as we are very eager to address them.
>
> Thank you very much!

---

### Decision · Program_Chairs · 2021-09-27

**Decision:**

Accept (Poster)

**Comment:**

This paper proposed a method which exploits principles of causal invariance from causality in order to develop an imitation learning algorithm which is more robust to spurious/distractor features in the expert trajectories. It shows the improved performance of this method in OpenAI Gym tasks and a healthcare dataset. The reviewers were unanimous in recommending acceptance, and no significant argument made by the lone weak accept has made me doubt this paper is worth publishing at NeurIPS. Without much difficulty, I can also recommend acceptance, and hope to see the minor improvements suggested by the reviewers in the final version of the paper.